# Discriminant Analysis of Anthropometric and Training Variables among Runners of Different Competitive Levels

**DOI:** 10.3390/ijerph18084248

**Published:** 2021-04-16

**Authors:** Mabliny Thuany, Raphael F. de Souza, Lee Hill, João Lino Mesquita, Thomas Rosemann, Beat Knechtle, Sara Pereira, Thayse Natacha Gomes

**Affiliations:** 1Centre of Research, Education, Innovation and Intervention in Sport (CIFI2D), Faculty of Sports, University of Porto, 4200-450 Porto, Portugal; mablinysantos@gmail.com (M.T.); joao_lino_mesquita@hotmail.com (J.L.M.); sara.s.p@hotmail.com (S.P.); 2Department of Physical Education, Federal University of Sergipe (UFS), São Cristóvão 49100-000, Brazil; raphaelctba20@hotmail.com (R.F.d.S.); thayse_natacha@hotmail.com (T.N.G.); 3Department of Pediatrics, Division of Gastroenterology and Nutrition, McMaster University, Hamilton, ON L8N 3Z5, Canada; hilll14@mcmaster.ca; 4Institute of Primary Care, University of Zurich, 8091 Zurich, Switzerland; thomas.rosemann@usz.ch; 5Medbase St. Gallen Am Vadianplatz, Vadianstrasse 26, 9001 St. Gallen, Switzerland; 6Centre of Research in Sport, Physical Education, Exercise and Health (CIDEFES), Lusophone University, 1749-024 Lisboa, Portugal

**Keywords:** runners, discriminant analysis, performance

## Abstract

The purpose of this study was to investigate the multivariate profile of different types of Brazilian runners and to identify the discriminant pattern of the distinct types of runners, as a runners’ ability to self-classify well. The sample comprised 1235 Brazilian runners of both sexes (492 women; 743 men), with a mean age of 37.94 ± 9.46 years. Individual characteristics were obtained through an online questionnaire: Sex, age, body height (m) and body mass (kg), socioeconomic status, and training information (i.e., self-classification, practice time, practice motivation, running pace, frequency and training volume/week). Multivariate analysis of variance was conducted by sex and the discriminant analysis was used to identify which among running pace, practice time, body mass index and volume/training could differentiate groups such as “professional athletes”, “amateur athletes” and “recreational athletes”. For both sexes, running pace was the variable that better discriminated the groups, followed by BMI and volume/week. The practice time is not a good indicator to differentiate runner’s types. In both sexes, semi-professional runners were those that better self-classify themselves, with amateur runners presenting the highest classification error. This information can be used to guide the long-term training, athlete’s selection programs, and to identify the strengths and weaknesses of athletes.

## 1. Introduction

There is no one size fits all strategy to determine sport performance, given that performance differs across modalities and specific abilities [1]. Performance is multi-factorial and the identification of variables that allow us to describe and differentiate an athlete’s athletic ability poses a unique challenge [2]. Recently, interest in these variables has grown among amateur and non-professional athletes, especially in activities that are practiced on a large scale [3,4]. Over the past 10 years there has been a growth of 57% in the number of runners participating in marathon and endurance events, with a notable decrease of the gender gap of participants [5]. In 2019, a total of 459,029 marathon finishers were recorded, with most of the events held in the United States (61.6%), the United Kingdom (10.7%) and Canada (10.0%) [6]. In addition, considering 19,614,975 marathon results from 2008–2018 across the globe, there was an increase in the number of participants from India, Portugal and Ireland, while the most representative countries were the United States (456,700), the United Kingdom (97,254) and Germany (86,032) [7]. It is interesting to note that the increase in participation has not resulted in faster marathon completion times. Over the last twenty years the average finish time has increased by approximately forty minutes [7]. However, this is a positive sign as it does indicate that more amateur, recreational and non-professional runners are participating in these endurance events.

In the Brazilian context, there are approximately 4 million non-professional runners [8] training in various capacities to improve their performance in organised competitions and events [9]. However, there is considerable variation within this group of athletes, specifically relating to the amount of time spent training and overall running performance [10]. In this context, running pace is one of the main variables used to differentiate athletes at the various levels of competition [11]. Running pace is determined by the time taken to cover one kilometre or mile, and is expressed as time per distance covered (min/km or mile) [12]. At the international and elite level, this index has been used as a cut-off point to stratify athletes into different competitive levels during an event or to determine eligibility to participate in the competition (e.g., six majors marathon) [13]. Running pace determination is particularly effective when implemented as an expected running pace (calculated to achieve a specific time) versus actual running pace (pace being achieved) in order to track performance whether in training or competition [14].

In addition to running pace calculations, various anthropometric components [15] and training variables [16] have been used to determine an athlete’s competitive level. For example, Thuany et al. [10] found, in a study with Brazilian runners, that amateur runners who completed the highest volume and running frequency/week were four times more likely to produce a higher performance compared to recreational runners. In marathoners, the training velocity and body fat explained approximately 44% of the variance in performance [17]. Interestingly, in male half-marathoners, practice time, training volume, sum of skin folds and body mass index (BMI) accounted for approximately 90% of performance variance [18]. Besides the relevance of these studies, generally, these attempts are usually focused on a univariate competitive profile, leading to the presentation of different classifications to describe/classify runners, such as “amateur runners”, “recreational runners”, “competitive level runners” [9,10].

Considering the relationship between variables of different characteristics, information about multivariate predictors of performance and a discriminant analysis, encompassing classes of runners, is necessary. Therefore, the purposes of this study are (1) to describe the multivariate profile of different types of Brazilian runners, (2) to identify the discriminant pattern of the distinct types of runners, and (3) to verify the runners’ ability to self-classify. This information may assist the guidance of long-term training, athlete’s selection programs and identify the strengths and weaknesses of athletes.

## 2. Materials and Methods

### 2.1. Design and Sample

The data came from the “Intrack” project (https://intrackproject.wixsite.com/website), a cross-sectional research project conducted to identify the predictors of running performance based on an ecological approach [19]. The sample comprised 1235 runners of both sexes (492 women; 743 men), with a mean age of 37.9 ± 9.4 years (ranging from 18 to 72 years), from the five Brazilian regions (Southeast = 453 (36.7%); Northeast = 441 (35.7%); South = 145 (11.7%); Midwest = 104 (8.4%); North = 89 (7.2%); Missing data = 3 (0.2%)). To be considered eligible for the study, runners should have answered the online questionnaire; those aged below 18 years, and those that did not answer all the mandatory questions from the applied questionnaire were excluded during data analysis. This study was conducted in accordance with the Declaration of Helsinki, and was approved by the Ethics Committee of the Federal University of Sergipe, Brazil (protocol n° 3.558.630).

### 2.2. Procedures and Data Collection

Information collected was self-reported through the questionnaire section “Profile characterization and associated factors for runner’s performance” [20]. The instrument was available through an online social media platform (Facebook, Instagram, WhatsApp). The study was conducted between September 2019 and March 2020. The following information was obtained:

#### 2.2.1. Individual Characteristics

Sex, age, body weight (kg), and height (m) were self-reported. Body mass index (BMI) was computed using the standardized formula (weight (kg)/height (m)^2^).

#### 2.2.2. Demographic Information

Educational level was dichotomized as “ungraduated” and “graduated”. The socioeconomic status (SES) was categorized based on the Brazilian minimum wage in 2019 [21] as “≤3 minimum wages” and “>3 minimum wages”. The state of residence was given by the runners, which allowed the identification of the regions where the states belong. This information was used for group characterization.

#### 2.2.3. Training Information

Running pace: expressed in minutes/km, was self-reported by runners, taking into account their preferred distance. Practice time: runners reported their practice time in months. Frequency: this was reported in counts (1–7 session/week), and was further dichotomized into “at least 3 sessions/week” and “more than 3 sessions/week”. Volume/week: the mean value was reported (in kilometres) for runners, considering the weekly amount.

#### 2.2.4. Self-Classification

Runners were invited to answer the following question: “Regarding the race, you consider yourself as”: Professional athlete (has some employment relationship with sports companies/racing clubs); Amateur athlete (has no employment relationship with sports companies/racing clubs, but seeks to improve performance and participation in competitions); Recreational athlete (has no competitive interest with road racing)”. This classification was used during analysis.

### 2.3. Statistical Analysis

Descriptive statistics are presented as mean, standard deviation (SD), frequencies (%). Univariate normality was tested for BMI, running pace, practice years, and volume/week, by self-classification groups. For graphical representation, variables were standardized (running pace was multiplied by −1). The presence of multivariate outliers was tested by the Mahalanobis distance. To identify profile differences between runners’ classes, multivariate analysis of variance was conducted by sex, and Pillai’s trace values were considered, given that variance and covariance homogeneity were not observed. Eta squared (n^2^) was used as a measurement of the effect size. The discriminant analysis [22] was used to identify variables (BMI, running pace, practice time and volume/week) that could differentiate groups of “professional athletes”, “amateur athletes” and “recreational athletes”. The software IBM SPSS Statistics (IBM Corp. Released 2016. IBM SPSS Statistics for Windows, Version 26.0. Armonk, NY: IBM Corp) was used for the analysis. Significance was accepted at *p* < 0.05.

## 3. Results

Descriptive information is presented in Table 1. More than 50% of runners classified themselves as “amateur runners” (68.5% and 77.4%, among women and men, respectively) for the studied sample. For both sexes, the runners self-classified as “professional athletes” were the youngest ones, and reported a higher frequency of training (i.e., more than three sessions/week). Regarding the socio-economic information, this last group also presented the highest frequency of “ungraduated” and as having “≤3 minimum wages” for both sexes. Further, the majority of runners self-classified as “recreational runners” with a training frequency ≤3/week, and high educational and economic levels.

Figure 1 displays the multivariate graphical profiles of recreational, amateur and semi-professional runners. In both sexes, the same pattern was observed for the variables used to differentiate the groups. In both, the practice time was the variable that, visually, presented the lowest discrepancies among runners.

In both sexes, the multivariate variance analysis identified differences for variables among runners’ self-classification (Table 2). For women, a macro analysis indicated that 7.2% of the total variance was explained by belonging to the group; while for men, this group effect explains ≈9%. A mid-level analysis indicated that, for both sexes, only the practice time did not differ between runners. Moreover, the Bonferroni post-hoc showed, at a micro-level, significant differences between recreational and amateur runners for BMI, volume/week, and running pace for women, while among men, these variables differed between all groups.

The discriminant analysis indicated that only the first function explained the group variance in both sexes. So, for women and men, this variance explanation is 95.4% and 96.1%, respectively. For both sexes, running pace was the variable that better discriminated the groups, followed by BMI and volume/week (Table 3). The practice time was not a good indicator to differentiate the types of runners.

Table 4 presents the runners’ reclassification based on the discriminant function result. Among women, 48.2% of amateur runners, 59% of recreational runners, and 75% of professional athletes were well classified, while among men, 58%, 70.2% and 70.6%, of the amateur, recreational, and professional runners, respectively, were well classified. Among the groups, the amateur runners presented the highest error classification.

## 4. Discussion

During the last decades, the number of running events has increased considerably worldwide [6]. Between the runners, it is possible to identify sub-groups with different perspectives in practice. However, to the best of our knowledge, there is no available information regarding the use of the discriminant analysis to differentiate Brazilian runners from different competitive levels based on anthropometric and training variables. However, this statistical approach has been primarily used to differentiate athletes across different sports, based on a large number of variables, such as physical performance, motor coordination [23], anthropometric and biomotor variables among elite female adolescents of different sports [24], characteristics of soccer athletes selected and non-selected [25], and physiological variables [26].

It is interesting to note that in the sports context, athletes are usually clustered based on their performance. In non-professional sports, especially among amateur running events, athletes need to present their self-classification according to their running pace [13], but there is not enough evidence as to whether non-professional runners are able to classify themselves well. Hence, this study provides new insights, specifically that (i) running pace was the variable that better discriminates the groups, followed by BMI, and volume/week; (ii) practice time was not a good indicator to differentiate runners into groups; (iii) professional athletes were those who better classified themselves; and (iv) in both sexes, the amateur group presented the highest error classification.

Previous studies indicated the predictive power of anthropometric and training variables for road running performance [27]. Among training variables, the running pace has received the most attention, mainly during training [28], and for pacing strategy during competition [29]. Pacing strategy refers to the energy distribution during a workout session [29], and is associated with the athlete competitive level, experience, and also the previous knowledge regarding the race (e.g., race distance and route, and mean time to complete), besides the athlete’s physiological capacity [30]. In addition, an inverse relationship can be observed between aerobic capacity and pace variation [31], and previous studies indicated that the oldest and fastest marathoners presented a more consistent pace than the youngest and slowest ones [31,32], and these factors can explain the result found in the present study, where it was observed that running pace was the best discriminant variable, and that professional athletes were those with the highest accuracy in their self-classification (75% among women and 70% among men). Elite athletes tend to present a stricter pace control during training and competition [33], which can be associated with this result.

Both BMI and volume/week have been used for researchers by self-reported information, and they demonstrate adequate accuracy [34,35]. In addition to its widespread use as a predictor of running performance [28,36,37], it was previously reported that these variables allowed for adequate discrimination between amateur and recreational runners [10], where amateur runners presented the highest probability of reaching a greater volume/week and also of presenting with the lowest BMI values. Furthermore, the discriminant power of these variables can be due to their association with running characteristic, and the fact that they interact with each other. For example, the association between practice time, training volume, BMI, and the sum of skinfolds explains 90.3% of the performance variance in non-professional half-marathoners [18], and the training running speed and body fat percentage explains 44% of performance in recreational marathoners [17].

In disagreement with previous studies [18], among the Brazilian runners studied, the practice time was not a good discriminant variable. In general, the distinct groups of runners are classified based on running distance and/or running race place (e.g., mountain, street, middle-distance, ultra-marathoners), practice time (e.g., “novice” or “experienced”), or goals to be achieved with the practice (e.g., “recreational”, “amateur”) [9,38,39]. However, the relationship between performance and experience is not direct or causal. It is has been well established that runners with more experience in the practice show better physiological indicators [40], running economy, and stride frequency to optimize energy expenditure [41]. However, we have speculated that practice time *per se* is not a good indicator to differentiate groups of runners, because other aspects (quality of training, competition participation) need to be associated with long-term training.

Given that nowadays, about 50–75% of runners use apps/devices to monitor their progress, having real-time feedback of variables for training control (distance, volume, running pace, stride frequency, heart rate) [42,43], it was not expected that amateur runners showed the highest error of classification, especially when compared against recreational runners. For both sexes, in the reclassification, the majority of the amateur runners should be classified as “recreational runners”. However, it is interesting to note that among females, 20% were reclassified as “professional athlete”. We considered the definition of “professional athlete” based on the Brazilian law 9.615/98 [44], which states that professional athletes are those who receive financial remuneration, with a work contract. Therefore, it is possible that, based on this definition, some non-professional runners classified themselves as “amateur”, because they may not have an official work contract, notwithstanding they could have a performance similar to their professional athlete peers.

There are some limitations in the present study. These include the use of an online questionnaire to obtain information, which is prone to errors. However, the questionnaire has been previously validated, and similar approaches have been successfully used in other studies [45,46,47]. Specifically, for the use of anthropometric information, such as weight and height, notwithstanding the possibility of bias by the use of self-reported instrument, the use of this strategy among adults was shown to be relevantly accurate, supporting its use in research. The options for runners to self-classify themselves could have led to a misunderstanding, but each question was presented with a short definition minimizing any misinterpretation.

However, the strengths of the study included the use of the statistical approach in non-professional runners. Further, we included the use of variables that are easily measured to differentiate a runner’s classification in running events, selection programs and training monitoring. Through these results, we suggested that charts or reference values of running pace and BMI can be determined in future studies, considering the runner’s classification (e.g., amateur, recreational and semi-professional).

## 5. Conclusions

The running pace is the best variable to differentiate runners with different competitive levels, followed by BMI and volume/week. Between sexes, amateur runners are those that present with a higher error of classification. In contrast, semi-professional athletes present the best self-classification. Future studies can investigate physiological aspects and physical fitness components to differentiate runners into different categories, or even to discriminate them based on their preferred running distance (i.e., 5 km, 10 km, half-marathon, marathon, and ultramarathon). This information can be used in race events by organizations, or by coaches and athletes for long-term training development and potentially for talent identification and selection.

## Figures and Tables

**Figure 1 ijerph-18-04248-f001:**
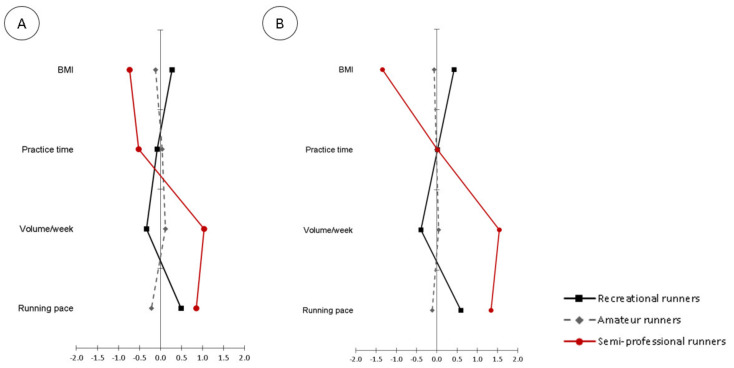
Multivariate profile for runners considering BMI, practice time, volume/week and running pace (**A**) Female runners; (**B**) Male runners. All variables were standardized.

**Table 1 ijerph-18-04248-t001:** Descriptive information for different types of runners, by sexes.

	Women			Men		
	Amateur Runners (*n* = 337)	Recreational Runners (*n* = 151)	Professional Athlete (*n* = 4)	Amateur Runners (*n* = 575)	Recreational Runners (*n* = 150)	Professional Athlete (*n* = 18)
Variables	Mean (SD) or Frequency (%)			Mean (SD) or Frequency (%)		
Age (years)	37.8 (8.5)	39.4 (9.0)	29.1 (12.2)	37.4 (39.8)	39.7 (10.5)	29.1 (9.3)
Regions						
Midwest	28 (8.3%)	18 (11.9%)	1 (25%)	45 (7.8%)	9 (6%)	3 (16.7%)
Northeast	99 (39.4%)	63 (41.7%)	1 (25%)	204 (35.5%	66 (44%)	8 (44.4%)
North	31 (9.2%)	8 (5.3%)	2 (50%)	44 (7.7%)	3 (2%)	1 (5.6%)
Southeast	128 (38%)	48 (31.8%)	0	218 (37.9%)	54 (36%)	5 (27.8%)
South	49 (14.5%)	14 (9.3%)	2 (50%)	63 (11%)	18 (12%)	1 (5.6%)
Frequency/Week						
≤3 train/week	222 (65.9%)	119 (78.8%)	1 (25%)	278 (48.3%)	113 (75.3%)	4 (22.2%)
>3 train/week	115 (34.1%)	32 (21.2%)	3 (75%)	297 (51.7%)	37 (24.7%)	14 (77.8%)
School Level						
Ungraduated	75 (22.3%)	24 (16%)	3 (75%)	195 (34.2%)	35 (23.6%)	15 (83.3%)
Graduated	262 (77.7%)	126 (84%)	1 (25%)	376 (65.8%)	113 (76.4%)	3 (16.7%)
SES						
≤3 minimum wage	104 (31.2%)	39 (26.4%)	2 (50%)	197 (34.7%)	49 (32.7%)	15 (83.3%)
>3 minimum wage	229 (68.8%)	109 (72.2%)	2 (50%)	371 (64.5%)	100 (66.7%)	3 (16.7%)

**Table 2 ijerph-18-04248-t002:** Results for multivariate analysis of variance for both sexes.

	Women					Men				
	Amateur Runners	Recreational Runners	Professional Athlete	*p*-Value	n^2^	Amateur Runners	Recreational Runners	Professional Athlete	*p*-Value	n^2^
	Mean (SD)					Mean (SD)				
BMI (kg/m^2^)	23.4 (3.2)	24.6 (2.9) *	21.47 (4.6)	<0.001	0.044	24.3 (2.7)	25.7 (2.7) *,†	20.6 (2.9)	<0.001	0.078
Practice time (months)	49.6 (37.5)	45.2 (35.9)	29 (16.0)	0.371	0.005	66.6 (69.3)	68.4 (71.1)	68.4 (47.0)	0.49	0.002
Volume/week (km)	30.0 (18.8)	22.2 (11.8) *	46.25 (32.0)	<0.001	0.054	42.8 (34.6)	28.1 (13.8) *,†	93.8 (58.4)	<0.001	0.092
Running pace (s)	354.7(56.1)	405.3 (88.1) *	308.25 (94.4)	<0.001	0.102	300.7 (53.8)	343.4 (67.4) *,†	226.3 (50.8)	<0.001	0.112
MANOVA Test	[(Pillai’s trace = 0.144); F (8,840) = 612.21, *p* < 0.001; n^2^ = 0.072]					[(Pillai’s trace = 0.182); F (8,1310) = 16.37, *p* < 0.001; n^2^ = 0.091]				

Note: * statistically different from amateur runners; † statistically different from professional athletes; n^2^ partial eta squared.

**Table 3 ijerph-18-04248-t003:** Function discriminant results, split by sex.

Women					Men			
	Wilks’ Ʌ	*p*-Value	Function		Wilks’ Ʌ	*p*-Value	Function	
Variables			1	2			1	2
BMI	0.956	<0.001	0.535	−0.013	0.922	<0.001	0.637	−0.102
Practice time	0.995	0.371	−0.055	0.812	0.998	0.490	0.080	0.293
Volume/week	0.946	<0.001	−0.597	−0.269	0.908	<0.001	−0.688	0.530
Running pace	0.898	<0.001	0.839	−0.390	0.888	<0.001	0.772	0.564

**Table 4 ijerph-18-04248-t004:** Runners’ reclassification based on function discriminant results.

	Amateur Runners	Recreational Runners	Professional Athletes	Total
Women				
Amateur runners	144 (48.2%)	95 (31.8%)	60 (20.1%)	299
Recreational runners	48 (39.3%)	72 (59%)	2 (1.6%)	122
Professional athletes	0 (0%)	1 (25%)	3 (75%)	4
Men				
Amateur runners	307 (58%)	167 (31.6%)	55 (10.4%)	529
Recreational runners	34 (29.8%)	80 (70.2%)	0 (0%)	114
Professional athletes	2 (11.8%)	3 (17.6%)	12 (70.6%)	17

## Data Availability

The data are not publicly available due to ethical concerns.

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
