# Peer review of "Discriminant Analysis of Anthropometric and Training Variables among Runners of Different Competitive Levels"

_ijerph, 2021, doi:10.3390/ijerph18084248_

Round 1

Reviewer 1 Report

This study attempted to examine the profile of different runners from Brazil and determine variables could explain the self-classification of these runners. The study was fairly well conducted with lots of ideas that can be generated for future research to expand upon the findings from the present study. Please see my comments below which as mainly grammatical.

Line 38: “these”

Line 42: “finishers”

Line 45: “has increased”

Line 58: delete “in”

Line 63: “to determine an athlete’s”

Line 64: “amateur runners”

Line 79: delete “well”

Line 79: “This information may assist with the guidance of long-term training….”

Line 114: “at least 3 sessions/week”

Line 118: delete “it”

Line 119: delete “it”

Line 121: delete “it”

Lines 134-135: “…that could differentiate groups of “professional athletes”, “amateur athletes” and “recreational athletes.”

Line 139: “Table 1”

Line 141: “professional athletes”

Lines 142-143: “…training (i.e. more than three sessions/week).

Line 152: “..was observed for the variables used to differentiate the groups.”

Line 153: “…the practice time was the variable that...”

Line 157: “(Table 2)”

Line 165: “…first function explained the group..”

Lines 174-192: Need to improve the presentation of Figure 1.

Line 236: “not enough evidence”

Line 237: “classify”

Line 243: “road running performance”

Line 245: “workout session”

Line 246: “..and is associated..”

Line 248: “…the athlete’s physiological capacity..”

Lines 248-250: This sentence is incomplete. Please revise.

Line 252: “…it was found that running pace was the best discriminant variable..”

Line 253: “professional athletes”

Line 261: For this sentence you have not identified any factors? Please revise to improve the clarity.

Line 268: “researchers”

Line 293: “similar approaches”

Line 294: “..options for runners self-classify themselves..”

Line 303: “…can be used in race events…”

Line 304: “..potentially for talent identification and selection.”

Line 307: “Between sexes, amateur runners are those that present with more error of..”

Lines 310-311: This part of the sentence is difficult to understand. Please revise.

Author Response

This study attempted to examine the profile of different runners from Brazil and determine variables could explain the self-classification of these runners. The study was fairly well conducted with lots of ideas that can be generated for future research to expand upon the findings from the present study. Please see my comments below which as mainly grammatical.

Authors’ answer: We thanks the reviewer for the comments and suggestions.

Line 38: “these”

Authors’ answer: Done

Line 42: “finishers”

Authors’ answer: Done

Line 45: “has increased”

Authors’ answer: Done

Line 58: delete “in”

Authors’ answer: Done

Line 63: “to determine an athlete’s”

Authors’ answer: Done

Line 64: “amateur runners”

Authors’ answer: Done

Line 79: delete “well”

Authors’ answer: Done

Line 79: “This information may assist with the guidance of long-term training….”

Authors’ answer: Done

Line 114: “at least 3 sessions/week”

Authors’ answer: Done

Line 118: delete “it”

Authors’ answer: Done

Line 119: delete “it”

Authors’ answer: Done

Line 121: delete “it”

Authors’ answer: Done

Lines 134-135: “…that could differentiate groups of “professional athletes”, “amateur athletes” and “recreational athletes.”

Authors’ answer: Done

Line 139: “Table 1”

Authors’ answer: Done

Line 141: “professional athletes”

Authors’ answer: Done

Lines 142-143: “…training (i.e. more than three sessions/week).

Authors’ answer: Done

Line 152: “..was observed for the variables used to differentiate the groups.”

Authors’ answer: Done

Line 153: “…the practice time was the variable that...”

Authors’ answer: Done

Line 157: “(Table 2)”

Authors’ answer: Done

Line 165: “…first function explained the group..”

Authors’ answer: Done

Lines 174-192: Need to improve the presentation of Figure 1.

Authors’ answer: Adjusted

Line 236: “not enough evidence”

Authors’ answer: Done

Line 237: “classify”

Authors’ answer: Done

Line 243: “road running performance”

Authors’ answer: Done

Line 245: “workout session”

Authors’ answer: Done

Line 246: “..and is associated..”

Authors’ answer: Done

Line 248: “…the athlete’s physiological capacity..”

Authors’ answer: Done

Lines 248-250: This sentence is incomplete. Please revise.

Authors’ answer: The sentence was rewrite.

Line 252: “…it was found that running pace was the best discriminant variable..”

Authors’ answer: Done

Line 253: “professional athletes”

Authors’ answer: Done

Line 261: For this sentence you have not identified any factors? Please revise to improve the clarity.

Authors’ answer: The sentence was rewrite.

Line 268: “researchers”

Authors’ answer: We maintained “researches”, since we were talking about “studies”.

Line 293: “similar approaches”

Authors’ answer: Done

Line 294: “..options for runners self-classify themselves..”

Authors’ answer: Done

Line 303: “…can be used in race events…”

Authors’ answer: Done

Line 304: “..potentially for talent identification and selection.”

Authors’ answer: Done

Line 307: “Between sexes, amateur runners are those that present with more error of..”

Authors’ answer: Done

Lines 310-311: This part of the sentence is difficult to understand. Please revise.

Authors’ answer: Done

Reviewer 2 Report

In the introduction there is an information of the marathons in the USA, Australia and New Zealand, but nothing of the mega marathon events organized in Europe (?)

Range of the runners age is very significant. Instead of segregating them in the context of Brazilian regions, for the scientific purpose I would suggest to divide those cohort onto the age groups i.e. 18-30; 31-49; 50-69 and on (according WHO’s age range proposition).

According BMI as one of the main study factors, in my opinion, the Authors’ approach to the computing the results from the online (declarative) data is not the best and appropriate way. Please, at least in the limitation give some necessary note to show your awareness of that matter.

Practice time would be easily collected by using a very good and well represented in the research practice (the level of PA a screening measure) proposed by Prochaska, Salis, and Long called Physical Activity Screening Measure, which is more adequate to compare in other research data from that field.

The statistics are very properly taken, this needs to be more complexed showing the relationship between the factors examined in the different assessed groups.      

Moreover I cannot find the Discussion section, there is no of limitations of the study.

From reviewer point of view I rate the prepared article as not ready for publication yet.

Author Response

In the introduction there is an information of the marathons in the USA, Australia and New Zealand, but nothing of the mega marathon events organized in Europe (?)

Authors’ answer: We appreciate the reviewer’s comment, and the sentence was rewritten.

Range of the runners age is very significant. Instead of segregating them in the context of Brazilian regions, for the scientific purpose I would suggest to divide those cohort onto the age groups i.e. 18-30; 31-49; 50-69 and on (according WHO’s age range proposition).

Authors’ answer: We appreciate the reviewer suggestion. However, please note that in running context, there is no consensus regarding age intervals used to classify runners in a given age category (this category is taking into account, in race events, to classify runners, per category, based on their performance), meaning that the use of age groups suggested may did not represent what is observed in running context. In addition, information regarding Brazilian regions were presented only with the purpose to characterize the sample, and this information was not used during data analysis (nor the age).

According BMI as one of the main study factors, in my opinion, the Authors’ approach to the computing the results from the online (declarative) data is not the best and appropriate way. Please, at least in the limitation give some necessary note to show your awareness of that matter.

Authors’ answer: We thank the reviewer for this observation, and we introduced this point in the limitation section. However, this strategy has been previously used in researches, and information related to agreement between two methods (measurement and self-reported) in runners was previously provided[1].

Practice time would be easily collected by using a very good and well represented in the research practice (the level of PA a screening measure) proposed by Prochaska, Salis, and Long called Physical Activity Screening Measure, which is more adequate to compare in other research data from that field.

Authors’ answer: We appreciate the reviewers’ suggestion. Please, note that we wanted to obtain information regarding runners practice time into the modality, and not regarding their involvement in “general” physical activity. Furthermore, in studies with runners, this information is usually presented as it was in our study, meaning that we may not have problems to compare our data with other research data from this field.

The statistics are very properly taken, this needs to be more complexed showing the relationship between the factors examined in the different assessed groups.      

Moreover, I cannot find the Discussion section, there is no of limitations of the study.

Authors’ answer: Done.

From reviewer point of view I rate the prepared article as not ready for publication yet.

Authors’ answer: We performed changes in the manuscript, which we believe will help to improve its quality.

[1] Nikolaidis, P.T.; Knechtle, B. Validity of recreational marathon runners' self-reported anthropometric data. Percept Mot Skills. 2020, 10.1177/0031512520930159, 31512520930159, doi:10.1177/0031512520930159.

Reviewer 3 Report

In general, the manuscript is well written except for the minor language corrections in the introduction section. The topic is very interesting for coaches and sports scientists to improve the training concepts. In addition, the novelty in this manuscript was the clearance of how we individualize the runners of competition formats to different maturation groups, dependent upon their physical development needs. 

Author Response

In general, the manuscript is well written except for the minor language corrections in the introduction section. The topic is very interesting for coaches and sports scientists to improve the training concepts. In addition, the novelty in this manuscript was the clearance of how we individualize the runners of competition formats to different maturation groups, dependent upon their physical development needs. 

Authors’ answer: We thank the reviewer for her/his comments. We performed the grammar check, to correct language mistakes/typos.

Reviewer 4 Report

Dear Authors,

thank you for the opportunity to review such an interesting text and learn about the research results.

I have the following comments: - there is no cancellation regarding the division of the respondents into: Professional athlete, Amateur athlete and Recreational athlete. What is the difference between amateur and recreational athlete?

- The "discussion" part is very extensive,

- after presenting and interpreting the results, we have "conclusions" section, which are very modest. I think there is a lot of contrast after the discussion.

Congratulations on such interesting research and good text. 

Author Response

Dear Authors,

thank you for the opportunity to review such an interesting text and learn about the research results.

I have the following comments: - there is no cancellation regarding the division of the respondents into: Professional athlete, Amateur athlete and Recreational athlete. What is the difference between amateur and recreational athlete?

Authors’ answer: Thanks for this question, and the opportunity to discuss about the classification we used in the manuscript. Definitions used by us were presented to runners, as the questionnaire answered, and also in the manuscript. In general, there is no consensus in the literature regarding the most appropriate term to be used to classify non-professional runners (or even practitioners from other modalities). If observed previous researches, a lot of classification are presented, such as “recreational runners”, “competitive runners”, “amateur runners”, “advanced runners” and “expert runners”. Trying to minimize this discrepancy, and with the ambitious goal to present a classification able to be generally used, we proposed, in a previous study[1] using the latent class analysis, the classification of non-professional runners into “amateur” and “recreational”. Based on our findings, the main differences between these two groups are related to “motivation for the practice” (amateur runners tend to be engaged into the practice with the purpose to increase the performance, while recreational runners aim to improve health and quality of life) and volume and number of training sections/week (amateur take part in more training sections/week). These aspects highlight that amateur runners have different goals into the practice, when compared to recreational ones, and also, they tend to have better performance, and these are the main differences between them.  

- The "discussion" part is very extensive,

Authors’ answer: We appreciate this observation, and we also agree with the reviewer. However, please note that given the extension of the results found, we tried to present possible explanation to them. We hope the reviewer understands our decision, and our explanation.

- after presenting and interpreting the results, we have "conclusions" section, which are very modest. I think there is a lot of contrast after the discussion.

Authors’ answer: Thanks for this observation. Indeed, our conclusion looked quite short. We tried to increase it.

Congratulations on such interesting research and good text. 

Authors’ answer: Thanks for the kindly comments and suggestions.

[1] Thuany, Mabliny, Gomes, Thayse Natacha, & Almeida, Marcos Bezerra de. (2020). Is there any difference between “amateur” and “recreational” runners? A latent class analysis. Motriz: Revista de Educação Física, 26(4), e10200140. Epub December 18, 2020.https://doi.org/10.1590/s1980-65742020000400140

Round 2

Reviewer 2 Report

Now I believe the paper is ready for publishing.

Best regards